# Rethinking the Evaluation for Conversational Recommendation in the Era of Large Language Models

**Xiaolei Wang**[1,3*], **Xinyu Tang**[1,3*], **Wayne Xin Zhao**[1,3†], **Jingyuan Wang**[4] and **Ji-Rong Wen**[1,2,3]

[1]Gaoling School of Artificial Intelligence, Renmin University of China
[2]School of Information, Renmin University of China
[3]Beijing Key Laboratory of Big Data Management and Analysis Methods
[4]School of Computer Science and Engineering, Beihang University
wxl1999@foxmail.com, txy20010310@163.com, batmanfly@gmail.com

## Abstract

The recent success of large language models (LLMs) has shown great potential to develop more powerful conversational recommender systems (CRSs), which rely on natural language conversations to satisfy user needs. In this paper, we embark on an investigation into the utilization of ChatGPT for CRSs, revealing the inadequacy of the existing evaluation protocol. It might overemphasize the matching with ground-truth items annotated by humans while neglecting the interactive nature of CRSs.

To overcome the limitation, we further propose an **i**nteractive **Eva**luation approach based on **LLM**s, named **iEvaLM**, which harnesses LLM-based user simulators. Our evaluation approach can simulate various system-user interaction scenarios. Through the experiments on two public CRS datasets, we demonstrate notable improvements compared to the prevailing evaluation protocol. Furthermore, we emphasize the evaluation of explainability, and ChatGPT showcases persuasive explanation generation for its recommendations. Our study contributes to a deeper comprehension of the untapped potential of LLMs for CRSs and provides a more flexible and realistic evaluation approach for future research about LLM-based CRSs. The code is available at https://github.com/RUCAIBox/iEvaLM-CRS.

## 1 Introduction

Conversational recommender systems (CRSs) aim to provide high-quality recommendation services through natural language conversations that span multiple rounds. Typically, in CRSs, *a recommender module* provides recommendations based on user preferences from the conversation context, and *a conversation module* generates responses given the conversation context and item recommendation.

---

*Equal contribution.
†Corresponding author.

Since CRSs rely on the ability to understand and generate natural language conversations, capable approaches for CRSs have been built on pre-trained language models in existing literature (Wang et al., 2022c; Deng et al., 2023). More recently, large language models (LLMs) (Zhao et al., 2023a), such as ChatGPT, have shown that they are capable of solving various natural language tasks via conversations. Since ChatGPT has acquired a wealth of world knowledge during pre-training and is also specially optimized for conversation, it is expected to be an excellent CRS. However, there still lacks a comprehensive study of how LLMs (*e.g.,* Chat-GPT) perform in conversational recommendation.

To investigate the capacity of LLMs in CRSs, we conduct an empirical study on the performance of ChatGPT on existing benchmark datasets. We follow the standard evaluation protocol and compare ChatGPT against state-of-the-art CRS methods. Surprisingly, the finding is rather *counter-intuitive*: ChatGPT shows unsatisfactory performance in this empirical evaluation. To comprehend the reason behind this discovery, we examine the failure cases and discover that the current evaluation protocol is the primary cause. It relies on the matching between manually annotated recommendations and conversations and might overemphasize the fitting of ground-truth items based on the conversation context. Since most CRS datasets are created in a chit-chat way, we find that these conversations are often vague about the user preference, making it difficult to exactly match the ground-truth items even for human annotation. In addition, the current evaluation protocol is based on fixed conversations, which does not take the interactive nature of conversational recommendation into account. Similar findings have also been discussed on text generation tasks (Bang et al., 2023; Qin et al., 2023): traditional metrics (*e.g.,* BLEU and ROUGE) may not reflect the real capacities of LLMs.

Considering this issue, we aim to improve the

evaluation approach, to make it more focused on the interactive capacities of CRSs. Ideally, such an evaluation approach should be conducted by humans, since the performance of CRSs would finally be tested by real users in practice. However, user studies are both expensive and time-consuming, making them infeasible for large-scale evaluations. As a surrogate, user simulators can be used for evaluation. However, existing simulation methods are typically limited to pre-defined conversation flows or template-based utterances (Lei et al., 2020; Zhang and Balog, 2020). To address these limitations, a more flexible user simulator that supports free-form interaction in CRSs is actually needed.

To this end, this work further proposes an **i**nteractive **Eva**luation approach based on **LLM**s, named **iEvaLM**, in which LLM-based user simulation is conducted to examine the performance. Our approach draws inspiration from the remarkable instruction-following capabilities exhibited by LLMs, which have already been leveraged for role-play (Fu et al., 2023). With elaborately designed instructions, LLMs can interact with users in a highly cooperative manner. Thus, we design our user simulators based on LLMs, which can flexibly adapt to different CRSs without further tuning. Our evaluation approach frees CRSs from the constraints of rigid, human-written conversation texts, allowing them to interact with users in a more natural manner, which is close to the experience of real users. To give a comprehensive evaluation, we also consider two types of interaction: attribute-based question answering and free-form chit-chat.

With this new evaluation approach, we observe significant improvements in the performance of ChatGPT, as demonstrated through assessments conducted on two publicly available CRS datasets. Notably, the Recall@10 metric has increased from 0.174 to 0.570 on the REDIAL dataset with five-round interaction, even surpassing the Recall@50 result of the currently leading CRS baseline. Moreover, in our evaluation approach, we have taken the crucial aspect of explainability into consideration, wherein ChatGPT exhibits proficiency in providing persuasive explanations for its recommendations. Besides, existing CRSs can also benefit from the interaction, which is an important ability overlooked by the traditional evaluation. However, they perform much worse in the setting of attribute-based question answering on the OPENDIALKG dataset, while ChatGPT performs better in both settings on

| Dataset | #Dialogues | #Utterances | Domains |
|---------|-----------|-------------|---------|
| ReDial | 10,006 | 182,150 | Movie |
| OpenDialKG | 13,802 | 91,209 | Movie, Book, Sports, Music |

Table 1: Statistics of the datasets.

the two datasets. It demonstrates the superiority of ChatGPT across different scenarios, which is expected for a general-purpose CRS.

We summarize our key contributions as follows:

(1) To the best of our knowledge, it is the first time that the capability of ChatGPT for conversational recommendation has been systematically examined on large-scale datasets.

(2) We provide a detailed analysis of the limitations of ChatGPT under the traditional evaluation protocol, discussing the root cause of why it fails on existing benchmarks.

(3) We propose a new interactive approach that employs LLM-based user simulators for evaluating CRSs. Through experiments conducted on two public CRS datasets, we demonstrate the effectiveness and reliability of our evaluation approach.

## 2 Background and Experimental Setup

In this section, we describe the task definition and experimental setup used in this work.

### 2.1 Task Description

Conversational Recommender Systems (CRSs) are designed to provide item recommendations through multi-turn interaction. The interaction can be divided into two main categories: question answering based on templates (Lei et al., 2020; Tu et al., 2022) and chit-chat based on natural language (Wang et al., 2023; Zhao et al., 2023c). In this work, we consider the second category. At each turn, the system either presents a recommendation or initiates a new round of conversation. This process continues until the user either accepts the recommended items or terminates the conversation. In general, CRSs consist of two major subtasks: recommendation and conversation. Given its demonstrated prowess in conversation (Zhao et al., 2023b), we focus our evaluation of ChatGPT on its performance in the recommendation subtask.

### 2.2 Experimental Setup

**Datasets.** We conduct experiments on the RE-DIAL (Li et al., 2018) and OPENDIALKG (Moon

et al., 2019) datasets. REDIAL is the most commonly used dataset in CRS, which is about movie recommendations. OPENDIALKG is a multi-domain CRS dataset covering not only movies but also books, sports, and music. Both datasets are widely used for CRS evaluation. The statistics for them are summarized in Table 1.

**Baselines.** We present a comparative analysis of ChatGPT with a selection of representative supervised and unsupervised methods:

• *KBRD* (Chen et al., 2019): It introduces DBpedia to enrich the semantic understanding of entities mentioned in dialogues.

• *KGSF* (Zhou et al., 2020): It leverages two KGs to enhance the semantic representations of words and entities and use Mutual Information Maximization to align these two semantic spaces.

• *CRFR* (Zhou et al., 2021a): It performs flexible fragment reasoning on KGs to address their inherent incompleteness.

• *BARCOR* (Wang et al., 2022b): It proposes a unified CRS based on BART (Lewis et al., 2020), which tackles two tasks using a single model.

• *MESE* (Yang et al., 2022): It formulates the recommendation task as a two-stage item retrieval process, *i.e.,* candidate selection and ranking, and introduces meta-information when encoding items.

• *UniCRS* (Wang et al., 2022c): It designs prompts with KGs for DialoGPT (Zhang et al., 2020) to tackle two tasks in a unified approach.

• *text-embedding-ada-002* (Neelakantan et al., 2022): It is a powerful model provided in the OpenAI API to transform each input into embeddings, which can be used for recommendation.

Among the above baselines, `text-embedding-ada-002` is an unsupervised method, while others are supervised and trained on CRS datasets.

**Evaluation Metrics.** Following existing work (Zhang et al., 2023; Zhou et al., 2022), we adopt Recall@$k$ to evaluate the recommendation subtask. Specifically, we set $k$ = 1, 10, 50 following Zhang et al. (2023) for the REDIAL dataset, and $k$ = 1, 10, 25 following Zhou et al. (2022) for the OPENDIALKG dataset. Since requiring too many items can sometimes be refused by ChatGPT, we only assess Recall@1 and Recall@10 for it.

**Model details.** We employ the publicly available model `gpt-3.5-turbo` provided in the OpenAI API, which is the underlying model of ChatGPT. To make the output as deterministic as possible, we

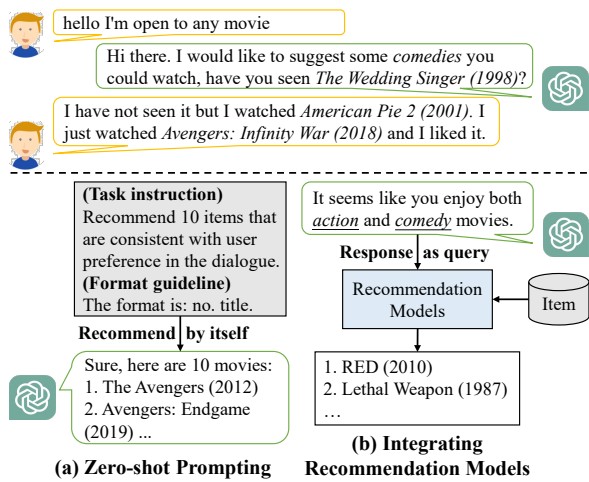

Figure 1: The method of adapting ChatGPT for CRSs.

set `temperature=0` when calling the API. All the prompts we used are detailed in Appendix C.

## 3 ChatGPT for Conversational Recommendation

In this section, we first discuss how to adapt ChatGPT for CRSs, and then analyze its performance.

### 3.1 Methodology

Since ChatGPT is specially optimized for dialogue, it possesses significant potential for conversational recommendation. Here we propose two approaches to stimulating this ability, as illustrated in Figure 1.

**Zero-shot Prompting.** We first investigate the ability of ChatGPT through zero-shot prompting (see Appendix C.1). The prompt consists of two parts: task instruction (describing the task) and format guideline (specifying the output format).

**Integrating Recommendation Models.** Although ChatGPT can directly generate the items, it is not specially optimized for recommendation (Kang et al., 2023; Dai et al., 2023). In addition, it tends to generate items that are outside the evaluation datasets, which makes it difficult to directly assess the predictions. To bridge this gap, we incorporate external recommendation models to constrain the output space. We concatenate the conversation history and generated responses as inputs for these models to directly predict target items or calculate the similarity with item candidates for matching. We select the CRS model MESE (Yang et al., 2022) as the supervised method (ChatGPT + text-embedding-ada-002) and the `text-embedding-ada-002` (Neelakantan et al.,

| Datasets | ReDial | | | OpenDialKG | | |
|---|---|---|---|---|---|---|
| Models | Recall@1 | Recall@10 | Recall@50 | Recall@1 | Recall@10 | Recall@25 |
| KBRD | 0.028 | 0.169 | 0.366 | 0.231 | 0.423 | 0.492 |
| KGSF | 0.039 | 0.183 | 0.378 | 0.119 | 0.436 | 0.523 |
| CRFR | 0.040 | 0.202 | 0.399 | 0.130 | 0.458 | 0.543 |
| BARCOR | 0.031 | 0.170 | 0.372 | **0.312** | 0.453 | 0.510 |
| UniCRS | 0.050 | 0.215 | 0.413 | 0.308 | 0.513 | 0.574 |
| MESE | **0.056*** | **0.256*** | **0.455*** | 0.279 | **0.592*** | **0.666*** |
| text-embedding-ada-002 | 0.025 | 0.140 | 0.250 | 0.279 | 0.519 | 0.571 |
| ChatGPT | 0.034 | 0.172 | – | 0.105 | 0.264 | – |
| + MESE | 0.036 | 0.195 | – | 0.240 | 0.508 | – |
| + text-embedding-ada-002 | 0.037 | 0.174 | – | 0.310 | 0.539 | – |

Table 2: Overall performance of existing CRSs and ChatGPT. Since requiring too many items at once can sometimes be refused by ChatGPT, we only assess Recall@1 and Recall@10 for it, while Recall@50 is marked as "–". Numbers marked with * indicate that the improvement is statistically significant compared with the best baseline (t-test with p-value < 0.05).

| Dataset | Irrelevant | Partially relevant | Highly relevant |
|---|---|---|---|
| ReDial | 8% | 20% | 72% |
| OpenDialKG | 20% | 16% | 64% |

Table 3: The relevance degree of the explanations generated by ChatGPT to the conversation context.

2022) model provided in the OpenAI API as the unsupervised method (ChatGPT + MESE).

## 3.2 Evaluation Results

We first compare the accuracy of ChatGPT with CRS baselines following existing work (Chen et al., 2019; Zhang et al., 2023). Then, to examine the inner working principles of ChatGPT, we showcase the explanations generated by it to assess its explainability as suggested by Guo et al. (2023).

### 3.2.1 Accuracy

The performance comparison of different methods for CRS is shown in Table 2. Surprisingly, ChatGPT does not perform as well as we expect. When using zero-shot prompting, ChatGPT only achieves average performance among these baselines and is far behind the top-performing methods. When integrating external recommendation models, its performance can be effectively improved. In particular, on the OPENDIALKG dataset, the performance gap is significantly reduced. It indicates that the responses generated by ChatGPT can help external models understand the user preference. However, there is still a noticeable performance gap on the REDIAL dataset.

### 3.2.2 Explainability

To better understand how ChatGPT conducts the recommendation, we require it to generate an explanation to examine the inner working principles. Then, we employ two annotators to judge the relevance degree (irrelevant, partially relevant, or highly relevant) of the explanation to the conversation context on 100 randomly sampled failure examples. The Cohen's Kappa between annotators is 0.77, indicating good agreement. The results in Table 3 indicate that ChatGPT can give highly relevant explanations in most of the cases. Here is one example:

> **[Conversation History]**
> User: Hi I want a movie like *Super Troopers (2001)*
> **[Label]** Police Academy (1984)
> **[Prediction of ChatGPT]** Beerfest (2006), The Other Guys (2010), Hot Fuzz (2007), . . .
> **[Explanation of ChatGPT]** . . . I have recommended movies that share *similar themes* of comedy, law enforcement, and absurdity. . . . Some of the movies on the list are also from the *same creators* or feature some of the *same actors* as Super Troopers. . . .

As we can see, ChatGPT understands the user preference and gives reasonable explanations, suggesting that it can be a good CRS. However, this contradicts its poor performance in accuracy. It motivates us to investigate the reasons for failure.

### 3.3 Why does ChatGPT Fail?

In this part, we analyze *why* does ChatGPT fail in terms of accuracy. Through inspecting the incor-

| [Conversation History] | [Conversation History] |
|---|---|

(a) Lack of explicit user preference     (b) Lack of proactive clarification

Figure 2: Two failure examples of ChatGPT for conversation recommendation.

rect recommendations (evaluated according to the annotated labels), we identify two main causes and detail them in the following part.

**Lack of Explicit User Preference.** The examples in this class typically have very short conversation turns, in which CRSs may be unable to collect sufficient evidence to accurately infer the user intention. Furthermore, the conversations are mainly collected in chit-chat form, making it vague to reflect the real user preference. To see this, we present an example in Figure 2(a). As we can see, the user does not provide any explicit information about the expected items, which is a common phenomenon as observed by Wang et al. (2022a). To verify this, we randomly sample 100 failure examples with less than three turns and invite two annotators to determine whether the user preference is ambiguous. Among them, 51% examples are annotated as ambiguous, and the rest 49% are considered clear, which confirms our speculation. The Cohen's Kappa between annotators is 0.75. Compared with existing models trained on CRS datasets, such an issue is actually more serious for ChatGPT, since it is not fine-tuned and makes prediction solely based on the dialogue context.

**Lack of Proactive Clarification.** A major limitation in evaluation is that it has to strictly follow existing conversation flows. However, in real-world scenarios, a CRS would propose proactive clarification when needed, which is not supported by existing evaluation protocols. To see this, we present an example in Figure 2(b), As we can see, the response in the dataset directly gives recommendations, while ChatGPT asks for detailed user preference. Since so many items fit the current require-

ment, it is reasonable to seek clarification before making a recommendation. However, such cases cannot be well handled in the existing evaluation protocol since no more user responses are available in this process. To verify this, we randomly sample 100 failure examples for two annotators to classify the responses generated by ChatGPT (clarification, recommendation, or chit-chat). We find that 36% of them are clarifications, 11% are chit-chat, and only 53% are recommendations, suggesting the importance of considering clarification in evaluation. The Cohen's Kappa between annotators is 0.81.

To summarize, there are two potential issues with the existing evaluation protocol: lack of explicit user preference and proactive clarification. Although conversation-level evaluation (Zhang and Balog, 2020) allows system-user interaction, it is limited to pre-defined conversation flows or template-based utterances (Lei et al., 2020; Zhang and Balog, 2020), failing to capture the intricacies and nuances of real-world conversations.

## 4 A New Evaluation Approach for CRSs

Considering the issues with the existing evaluation protocol, in this section, we propose an alternative evaluation approach, **iEvaLM**, which features interactive evaluation with LLM-based user simulation, as illustrated in Figure 3. We demonstrate its effectiveness and reliability through experiments.

### 4.1 Overview

Our approach is seamlessly integrated with existing CRS datasets. Each system-user interaction extends over one of the observed human-annotated conversations. The key idea of our approach is to conduct close-to-real user simulation based on the

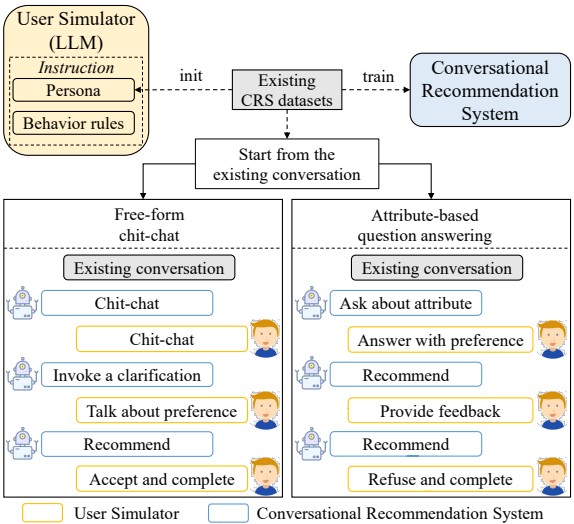

Figure 3: Our evaluation approach **iEvaLM**. It is based on existing CRS datasets and has two settings: free-form chit-chat (left) and attribute-based question answering (right).

excellent *role-play* capacities of LLMs (Fu et al., 2023). We take the ground-truth items as the user preference and use them to set up the persona of the LLM-based simulated user via instructions. After the interaction, we assess not only the accuracy by comparing predictions with the ground-truth items but also the explainability by querying an LLM-based scorer with the generated explanations.

## 4.2 Interaction Forms

To make a comprehensive evaluation, we consider two types of interaction: *attribute-based question answering* and *free-form chit-chat*.

In the first type, the action of the system is restricted to choosing one of the $k$ pre-defined attributes to ask the user or making recommendations. At each round, we first let the system decide on these $k + 1$ options, and then the user gives the template-based response: answering questions with the attributes of the target item or giving feedback on recommendations. An example interaction round would be like: "*System: Which genre do you like? User: Sci-fi and action.*"

In contrast, the second type does not impose any restrictions on the interaction, and both the system and user are free to take the initiative. An example interaction round would be like: "*System: Do you have any specific genre in mind? User: I'm looking for something action-packed with a lot of special effects.*"

## 4.3 User Simulation

To support the interaction with the system, we employ LLMs for user simulation. The simulated user can take on one of the following three behaviors:

- *Talking about preference.* When the system makes a clarification or elicitation about user preference, the simulated user would respond with the information about the target item.

- *Providing feedback.* When the system recommends an item list, the simulated user would check each item and provide positive feedback if finding the target or negative feedback if not.

- *Completing the conversation.* If one of the target items is recommended by the system or the interaction reaches a certain number of rounds, the simulated user would finish the conversation.

Specifically, we use the ground-truth items from existing datasets to construct realistic personas for simulated users. This is achieved by leveraging the `text-davinci-003` (Ouyang et al., 2022) model provided in the OpenAI API, which demonstrates superior instruction following capacity as an evaluator (Xu et al., 2023; Li et al., 2023). To adapt `text-davinci-003` for user simulation, we set its behaviors through manual instructions (see Appendix C.3). In these instructions, we first fill the ground-truth items into the persona template and then define their behaviors using a set of manually crafted rules. At each turn, we append the conversation to the instruction as input. When calling the API, we set `max_tokens` to 128, `temperature` to 0, and leave other parameters at their default values. The maximum number of interaction rounds is set to 5.

## 4.4 Performance Measurement

We consider both subjective and objective metrics to measure the recommendation performance as well as the user experience. For the objective metric, we use *recall* as stated in Section 2.2 to evaluate every recommendation action in the interaction process. For the subjective metric, following Chen et al. (2022), we use *persuasiveness* to assess the quality of explanations for the last recommendation action in the interaction process, aiming to evaluate whether the user can be persuaded to accept recommendations. The value range of this metric is {0, 1, 2}. To reduce the need for humans, we propose an LLM-based scorer that can automatically give the score through prompting. Specifically, we use the `text-davinci-003` (Ouyang et al., 2022) model

| Setting | Single-turn | | Multi-turn | |
|---|---|---|---|---|
| | Naturalness | Usefulness | Naturalness | Usefulness |
| DialoGPT | 13% | 23% | 11% | 31% |
| iEvaLM | **36%** | **43%** | **55%** | **38%** |
| Tie | 51% | 34% | 34% | 31% |
| Human | 10% | 34% | 17% | 28% |
| iEvaLM | **39%** | 33% | **35%** | **40%** |
| Tie | 51% | 33% | 48% | 32% |

Table 4: Performance comparison in terms of naturalness and usefulness in the single-turn and multi-turn settings. Each value represents the percentage of pairwise comparisons that the specific model wins or ties.

provided in the OpenAI API as the scorer with the conversation, explanation, and scoring rules concatenated as prompts (see Appendix C.4). Other parameters remain the same as the simulated user.

## 5 Evaluation Results

In this section, we assess the quality of the user simulator and the performance of CRSs using our proposed evaluation approach.

### 5.1 The Quality of User Simulator

To evaluate the performance of CRSs in an interactive setting, we construct user simulators based on ground-truth items from existing datasets. The simulated users should cooperate with the system to find the target item, *e.g.,* answer clarification questions and provide feedback on recommendations. However, it is not easy to directly evaluate the quality of user simulators.

Our solution is to make use of the annotated conversations in existing datasets. We first use the ground-truth items to set up the persona of the user simulator and then let them interact with the systems played by humans. They are provided with the first round of annotated conversations to complete the rest. Then, we can compare the completed conversations with the annotated ones for evaluation. Following Sekulić et al. (2022), we assess the *naturalness* and *usefulness* of the generated utterances in the settings of single-turn and multi-turn free-form chit-chat. Naturalness means that the utterances are fluent and likely to be generated by humans, and usefulness means that the utterances are consistent with the user preference. We compare our user simulator with a fine-tuned version of DialoGPT and the original conversations in the REDIAL dataset.

Specifically, we first invite five annotators to play the role of the system and engage in interac-

tions with each user simulator. The interactions are based on the first round of conversations from 100 randomly sampled examples. Then, we employ another two annotators to make pairwise evaluations, where one is generated by our simulator and the other comes from DialoGPT or the dataset. We count a win for a method when both annotators agree that its utterance is better; otherwise, we count a tie. The Cohen's Kappa between annotators is 0.73. Table 4 demonstrates the results. We can see that our simulator significantly outperforms DialoGPT, especially in terms of naturalness in the multi-turn setting, which demonstrates the strong language generation capability of LLMs. Furthermore, the usefulness of our simulator is better than others, indicating that it can provide helpful information to cooperate with the system.

### 5.2 The Performance of CRS

In this part, we compare the performance of existing CRSs and ChatGPT using different evaluation approaches. For ChatGPT, we use ChatGPT + text-embedding-ada-002 due to its superior performance in traditional evaluation (see Appendix C.2).

#### 5.2.1 Main Results

The evaluation results are presented in Table 5 and Table 6. Overall, most models demonstrate improved accuracy and explainability compared to the traditional approach. Among existing CRSs, the order of performance is *UniCRS > BARCOR > KBRD*. Both UniCRS and BARCOR utilize pre-trained models to enhance conversation abilities. Additionally, UniCRS incorporates KGs into prompts to enrich entity semantics for better understanding user preferences. It indicates that existing CRSs have the ability to interact with users for better recommendations and user experience, which is an important aspect overlooked in the traditional evaluation.

For ChatGPT, there is a significant performance improvement in both Recall and Persuasiveness, and the Recall@10 value even surpassing the Recall@25 or Recall@50 value of most CRSs on the two datasets. This indicates that ChatGPT has superior interaction abilities compared with existing CRSs and can provide high-quality and persuasive recommendations with sufficient information about the user preference. The results demonstrate the effectiveness of iEvaLM in evaluating the accuracy and explainability of recommendations for CRSs, especially those developed with LLMs.

| Model | KBRD | | | BARCOR | | | UniCRS | | | ChatGPT | | |
|---|---|---|---|---|---|---|---|---|---|---|---|---|
| Evaluation Approach | Original | iEvaLM (attr) | iEvaLM (free) | Original | iEvaLM (attr) | iEvaLM (free) | Original | iEvaLM (attr) | iEvaLM (free) | Original | iEvaLM (attr) | iEvaLM (free) |
| ReDial R@1 | 0.028 | 0.039 (+39.3%) | 0.035 (+25.0%) | 0.031 | 0.034 (+9.7%) | 0.034 (+9.7%) | 0.050 | 0.053 (+6.0%) | 0.107 (+114.0%) | 0.037 | **0.191*** (+416.2%) | 0.146 (+294.6%) |
| ReDial R@10 | 0.169 | 0.196 (+16.0%) | 0.198 (+17.2%) | 0.170 | 0.201 (+18.2%) | 0.190 (+11.8%) | 0.215 | 0.238 (+10.7%) | 0.317 (+47.4%) | 0.174 | **0.536*** (+208.0%) | 0.440 (+152.9%) |
| ReDial R@50 | 0.366 | 0.436 (+19.1%) | 0.453 (+23.8%) | 0.372 | 0.427 (+14.8%) | 0.467 (+25.5%) | 0.413 | 0.520 (+25.9%) | **0.602*** (+45.8%) | – | – | – |
| OpenDialKG R@1 | 0.231 | 0.131 (-43.3%) | 0.234 (+1.3%) | 0.312 | 0.264 (-15.4%) | 0.314 (+0.6%) | 0.308 | 0.180 (-41.6%) | 0.314 (+1.9%) | 0.310 | 0.299 (-3.5%) | **0.400*** (+29.0%) |
| OpenDialKG R@10 | 0.423 | 0.293 (-30.7%) | 0.431 (+1.9%) | 0.453 | 0.423 (-6.7%) | 0.458 (+1.1%) | 0.513 | 0.393 (-23.4%) | 0.538 (+4.9%) | 0.539 | 0.604 (+12.1%) | **0.715*** (+32.7%) |
| OpenDialKG R@25 | 0.492 | 0.377 (-23.4%) | 0.509 (+3.5%) | 0.510 | 0.482 (-5.5%) | 0.530 (+3.9%) | 0.574 | 0.458 (-20.2%) | **0.609*** (+6.1%) | – | – | – |

Table 5: Performance of CRSs and ChatGPT under different evaluation approaches, where "attr" denotes attribute-based question answering and "free" denotes free-form chit-chat. "R@$k$" refers to Recall@$k$. Since requiring too many items can sometimes be refused by ChatGPT, we only assess Recall@1 and 10 for it, while Recall@50 is marked as "–". Numbers marked with * indicate that the improvement is statistically significant compared with the rest methods (t-test with p-value < 0.05).

| Model | Evaluation Approach | ReDial | OpenDialKG |
|---|---|---|---|
| KBRD | Original | 0.638 | 0.824 |
| | iEvaLM | 0.766 (+20.1%) | 0.862 (+4.6%) |
| BARCOR | Original | 0.667 | 1.149 |
| | iEvaLM | 0.795 (+19.2%) | 1.211 (+5.4%) |
| UniCRS | Original | 0.685 | 1.128 |
| | iEvaLM | 1.015 (+48.2%) | 1.314 (+16.5%) |
| ChatGPT | Original | 0.787 | 1.221 |
| | iEvaLM | **1.331*** (+69.1%) | **1.513*** (+23.9%) |

Table 6: The persuasiveness of explanations. We only consider the setting of free-form chit-chat in iEvaLM. Numbers marked with * indicate that the improvement is statistically significant compared with the rest methods (t-test with p-value < 0.05).

| Method | Unpersuasive | Partially persuasive | Highly persuasive |
|---|---|---|---|
| iEvaLM | 1% | 5% | 94% |
| Human | 4% | 7% | 89% |

Table 7: The score distribution of persuasiveness ("unpersuasive" for 0, "partially persuasive" for 1, and "highly persuasive" for 2) from our LLM-based scorer and human on a random selection of 100 examples from the REDIAL dataset.

Comparing the two interaction settings, ChatGPT has demonstrated greater potential as a general-purpose CRS. Existing CRSs perform much worse in the setting of attribute-based question answering than in the traditional setting on the OPENDIALKG dataset. One possible reason is that they are trained on datasets with natural language conversations, which is inconsistent with the setting of attribute-based question answering. In contrast, ChatGPT performs much better in both settings on the two datasets, since it has been specially trained on conversational data. The results indicate the limitations of the traditional evaluation, which focuses only on a single conversation scenario, while our evaluation approach allows for a more holistic assessment of CRSs, providing valuable insights into their strengths and weaknesses across different types of interactions.

### 5.2.2 The Reliability of Evaluation

Recall that LLMs are utilized in the user simulation and performance measurement parts of iEvaLM as alternatives for humans in Section 4. Considering that the generation of LLMs can be unstable, in this part, we conduct experiments to assess the reliability of the evaluation results compared with using human annotators.

First, recall that we introduce the subjective metric *persuasiveness* for evaluating explanations in Section 4.4. This metric usually requires human evaluation, and we propose an LLM-based scorer as an alternative. Here we evaluate the reliability of our LLM-based scorer by comparing with human annotators. We randomly sample 100 examples with the explanations generated by ChatGPT and ask our scorer and two annotators to rate them separately with the same instruction (see Appendix C). The Cohen's Kappa between annotators is 0.83. Table 7 demonstrates that the two score distributions are similar, indicating the reliability of our LLM-

| Evaluation Approach | | KBRD | BARCOR | UniCRS | ChatGPT |
|---|---|---|---|---|---|
| iEvaLM | Recall@10 | 0.180 | 0.210 | 0.330 | 0.460 |
| | Persuasiveness | 0.810 | 0.860 | 1.050 | 1.330 |
| Human | Recall@10 | 0.210 | 0.250 | 0.370 | 0.560 |
| | Persuasiveness | 0.870 | 0.930 | 1.120 | 1.370 |

Table 8: The evaluation results using simulated and real users on a random selection of 100 examples from the REDIAL dataset.

based scorer as a substitute for human evaluators.

Then, since we propose an LLM-based user simulator as a replacement for humans to interact with CRSs, we examine the correlation between the values of metrics when using real vs. simulated users. Following Section 4.3, both real and simulated users receive the same instruction (see Appendix C) to establish their personas based on the ground-truth items. Each user can interact with different CRSs for five rounds. We randomly select 100 instances and employ five annotators and our user simulator to engage in free-form chit-chat with different CRSs. The results are shown in Table 8. We can see that the ranking obtained from our user simulator is consistent with that of real users, and the absolute scores are also comparable. It suggests that our LLM-based user simulator is capable of providing convincing evaluation results and serves as a reliable alternative to human evaluators.

## 6 Conclusion

In this paper, we systematically examine the capability of ChatGPT for conversational recommendation on existing benchmark datasets and propose an alternative evaluation approach, **iEvaLM**. First, we show that the performance of ChatGPT was unsatisfactory. Through analysis of failure cases, the root cause is the existing evaluation protocol, which overly emphasizes the fitting of ground-truth items based on conversation context. To address this issue, we propose an interactive evaluation approach using LLM-based user simulators.

Through experiments with this new approach, we have the following findings: (1) ChatGPT is powerful and becomes much better in our evaluation than the currently leading CRSs in both accuracy and explainability; (2) Existing CRSs also get improved from the interaction, which is an important aspect overlooked by the traditional evaluation; and (3) ChatGPT shows great potential as a general-purpose CRS under different settings and datasets. We also demonstrate the effectiveness and reliability of our evaluation approach.

Overall, our work contributes to the understanding and evaluation of LLMs such as ChatGPT for conversational recommendation, paving the way for further research in this field in the era of LLMs.

## Limitations

A major limitation of this work is the design of prompts for ChatGPT and LLM-based user simulators. We manually write several prompt candidates and select the one with the best performance on some representative examples due to the cost of calling model APIs. More effective prompting strategies like chain-of-thought can be explored for better performance, and the robustness of the evaluation framework to different prompts remains to be assessed.

In addition, our evaluation framework primarily focuses on the accuracy and explainability of recommendations, but it may not fully capture potential issues related to fairness, bias, or privacy concerns. Future work should explore ways to incorporate these aspects into the evaluation process to ensure the responsible deployment of CRSs.

## Acknowledgements

This work was partially supported by National Natural Science Foundation of China under Grant No. 62222215 and 72222022, Beijing Natural Science Foundation under Grant No. 4222027, and the Outstanding Innovative Talents Cultivation Funded Programs 2022 of Renmin University of China. Xin Zhao is the corresponding author.

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

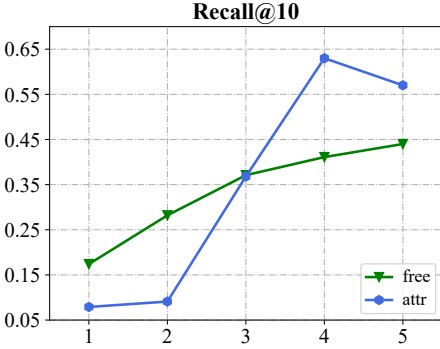

Figure 4: The performance of ChatGPT with different interaction rounds under the setting of attribute-based question answering (attr) and free-form chit-chat (free) on the REDIAL dataset.

## A The Influence of the Number of Interaction Rounds in iEvaLM

Interacting with the user for multiple rounds typically leads to more information and improved recommendation accuracy. However, users have limited patience and may leave the interaction when they become exhausted. It is important to investigate the relationship between the number of interaction rounds and performance. Following the setting in our approach, the interaction between ChatGPT and users is start from the observed human-annotated conversation in each dataset example, and we set the maximum interaction rounds to values from {1, 2, 3, 4, 5}, in order to evaluate the changes in recommendation accuracy.

Figure 4 shows the results of Recall@10 on the REDIAL dataset. In attribute-based question answering, the performance keeps increasing and reaches saturation at round 4. This observation aligns with our conversation setting, since the RE-DIAL dataset only has three attributes to inquire about. In free-form chit-chat, the performance curve is steep between rounds 1 and 3, while it is relatively flat between rounds 3 and 5. This pattern may be attributed to insufficient information in the initial round and marginal information in the last rounds. Since the user will gradually get exhausted with the progress of the interaction, how to optimize the conversation strategy remains to be further studied.

## B Related Work

In this section, we summarize the related work from the following perspectives.

### B.1 Conversational Recommender System

The fields of conversation intelligence (Chen et al., 2017; Gao et al., 2018) and recommendation systems (Wu et al., 2022) have seen significant progress in recent years. One promising development is the integration of these two fields, leading to the emergence of conversational recommender systems (CRSs) (Jannach et al., 2021; Gao et al., 2021). CRSs provide recommendations to users through conversational interactions, which has the potential to significantly improve the user experience.

One popular approach (Lei et al., 2020; Tu et al., 2022) assumes that interactions with users primarily take the form of question answering, where users are asked about their preferences for items and their attributes. The goal is to learn an optimal interaction strategy that captures user preferences and provides accurate recommendations in as few turns as possible. However, this approach often relies on hand-crafted templates and does not explicitly model the language aspect of CRSs. Another approach (Zhang et al., 2023; Zhou et al., 2022) focuses on engaging users in more free-form natural language conversations, such as chit-chat. The aim is to capture user preferences from the conversation context and generate recommendations using persuasive responses.

Our work belongs to the second category. In this work, we systematically evaluate the performance of large language models (LLMs) like ChatGPT for conversational recommendation on large-scale datasets.

### B.2 Language Models for Conversational Recommendation

There have been recent studies on how to integrate language models (LMs) into CRSs. One notable investigation by Penha and Hauff (2020) evaluates the performance of the pre-trained language model (PLM) BERT (Kenton and Toutanova, 2019) in conversational recommendation. Other studies (Wang et al., 2022c; Yang et al., 2022; Deng et al., 2023) primarily utilize PLMs as the foundation to build unified CRSs, capable of performing various tasks using a single model instead of multiple components. However, the current ap-

proaches are mainly confined to small-size LMs like BERT (Kenton and Toutanova, 2019) and DialoGPT (Zhang et al., 2020).

In this paper, we focus on the evaluation of CRSs developed with not only PLMs but also LLMs and propose a new evaluation approach iEvaLM.

## B.3 Evaluation and User Simulation

The evaluation of CRSs remains an area that has not been thoroughly explored in existing literature. Previous studies have primarily focused on turn-level evaluation (Chen et al., 2019), where the system output of a single turn is compared against ground-truth labels for two major tasks: conversation and recommendation. Some researchers have also adopted conversation-level evaluation to assess conversation strategies (Lei et al., 2020; Zhang et al., 2018; Balog and Zhai, 2023; Afzali et al., 2023). In such cases, user simulation is often employed as a substitute for human evaluation. These approaches typically involve collecting real user interaction history (Lei et al., 2020) or reviews (Zhang et al., 2018) to represent the preferences of simulated users. Zhou et al. (2021b) develop an open-source toolkit called CRSLab, which provides extensive and standard evaluation protocols. However, due to the intricate and interactive nature of conversational recommendation, the evaluation is often constrained by pre-defined conversation flows or template-based utterances. Consequently, this limitation hinders the comprehensive assessment of the practical utility of CRSs.

In our work, we propose an interactive evaluation approach iEvaLM with LLM-based user simulators, which has a strong instruction-following ability and can flexibly adapt to different CRSs based on the instruction without further tuning.

## C Prompts Used in the Paper

### C.1 Prompts for ChatGPT in the Traditional Evaluation

We use the following prompts for zero-shot prompting in section 3.1.

- ReDial

> Recommend 10 items that are consistent with user preference. The recommendation list can contain items that the dialog mentioned before. The format of the recommendation list is: no. title (year). Don't mention anything other than the title of items in your recommendation list.

- OpenDialKG

> Recommend 10 items that are consistent with user preference. The recommendation list can contain items that the dialog mentioned before. The format of the recommendation list is: no. title. Don't mention anything other than the title of items in your recommendation list.

### C.2 Prompts for ChatGPT in iEvaLM

We use the following prompts for ChatGPT in our new evaluation approach.

### C.2.1 Recommendation

**Free-Form Chit-Chat.**

- ReDial

> You are a recommender chatting with the user to provide recommendation. You must follow the instructions below during chat.
> If you do not have enough information about user preference, you should ask the user for his preference.
> If you have enough information about user preference, you can give recommendation. The recommendation list must contain 10 items that are consistent with user preference. The recommendation list can contain items that the dialog mentioned before. The format of the recommendation list is: no. title (year). Don't mention anything other than the title of items in your recommendation list.

- OpenDialKG

> You are a recommender chatting with the user to provide recommendation. You must follow the instructions below during chat.
> If you do not have enough information about user preference, you should ask the user for his preference.
> If you have enough information about user preference, you can give recommendation. The recommendation list must contain 10 items that are consistent with user preference. The recommendation list can contain items that the dialog mentioned before. The format of the recommendation list is: no. title. Don't mention anything other than the title of items in your recommendation list.

**Attribute-Based Question Answering.** "{}" refers to the options that have been selected.

- ReDial

> To recommend me items that I will accept, you can choose one of the following options.
> A: ask my preference for genre
> B: ask my preference for actor
> C: ask my preference for director
> D: I can directly give recommendations
> You have selected {}, do not repeat them.
> Please enter the option character.

- OpenDialKG

> To recommend me items that I will accept, you can choose one of the following options.
> A: ask my preference for genre
> B: ask my preference for actor
> C: ask my preference for director
> D: ask my preference for writer
> E: I can directly give recommendations
> You have selected {}, do not repeat them.
> Please enter the option character.

#### C.2.2 Explainability

> Please explain your last time of recommendation.

### C.3 Prompts for the User Simulator in iEvaLM

We use the following prompts for `text-davinci-003` to play the role of the user during interaction.

**Free-Form Chit-Chat.** "{}" refers to the item labels of each example in the datasets.

> You are a seeker chatting with a recommender for recommendation. Your target items: {}. You must follow the instructions below during chat.
> If the recommender recommends {}, you should accept.
> If the recommender recommends other items, you should refuse them and provide the information about {}. You should never directly tell the target item title.
> If the recommender asks for your preference, you should provide the information about {}. You should never directly tell the target item title.

### Attribute-Based Question Answering.

- When the recommended item list contains at least one of the target items:

> That's perfect, thank you!

- When the recommended item list does not contain any target item:

> I don't like them.

- When the system asks about the preference over pre-defined attributes, we use the attributes of target items as the answer if they exist, otherwise:

> Sorry, no information about this.

### C.4 Prompts for the LLM-based Scorer in iEvaLM

We use the following prompts for `text-davinci-003` to score the persuasiveness of explanations. "{}" refers to the item labels of each example in the datasets.

> Does the explanation make you want to accept the recommendation? Please give your score.
> If mention one of [{}], give 2.
> Else if you think recommended items are worse than [{}], give 0.
> Else if you think recommended items are comparable to [{}] according to the explanation, give 1.
> Else if you think recommended items are better than [{}] according to the explanation, give 2.
> Only answer the score number.