# OpenReview forum: "Rethinking the Evaluation for Conversational Recommendation in the Era of Large Language Models"
_EMNLP/2023/Conference — EMNLP 2023 Main_

### Official Review · Reviewer_ANzj · 2023-08-04

**Paper Topic And Main Contributions:** 1. This work is the first to systemat…
**Soundness:** 4

**Excitement:**

4: Strong: This paper deepens the understanding of some phenomenon or lowers the barriers to an existing research direction.

**Reasons To Accept:**

1.	The authors thoroughly analyse the current issues and challenges, and presents a novel approach to address them.

2.	This paper goes beyond theoretical discussions, providing thorough validation through detailed experiments and case studies.

3.	This paper is well-written in clear and concise language, demonstrating excellent organization and structure.

**Reasons To Reject:**

1.	You can use a bar chart as an alternative to a performance table.

2.	Taking into account the grammatical concern, when referring to "objective metrics," it's important to note that the term "objective" refers to a singular metric. Likewise, the same consideration applies to "subjective" metrics. Furthermore, you need to provide additional information about subjective metrics, such as how its formula is defined.

**Reproducibility:**

4: Could mostly reproduce the results, but there may be some variation because of sample variance or minor variations in their interpretation of the protocol or method.

**Reviewer Confidence:**

4: Quite sure. I tried to check the important points carefully. It's unlikely, though conceivable, that I missed something that should affect my ratings.

---

> ### Author Rebuttal · Authors · 2023-08-29
>
> Thank you for your valuable feedback on our paper. We appreciate the time and effort you have put into reviewing our work. We have carefully considered your comments and have addressed each of your concerns below.
>
> **1. You can use a bar chart as an alternative to a performance table.**
>
> Thanks for the suggestion! We use the performance table because it offers a comprehensive and detailed view of the results. However, we acknowledge your great suggestion of using a bar chart as an alternative to the performance table. It can provide a visual representation of the performance metrics, enhancing the visualization of our results. We will use the bar chart instead in the revised version.
>
> **2. Taking into account the grammatical concern, when referring to "objective metrics," it's important to note that the term "objective" refers to a singular metric. Likewise, the same consideration applies to "subjective" metrics. Furthermore, you need to provide additional information about subjective metrics, such as how its formula is defined.**
>
> Thanks for your careful reading! We apologize for the grammatical error and will rectify it in the revised version. Regarding the subjective metric persuasiveness, we have provided a detailed explanation of its definition, implementation, and scoring rules in Section 4.4 (Lines 430-443) and Appendix D.4. Since it is a simple and subjective metric (e.g., "if you think recommended items are worse than the ground-truth items, give 0" in Appendix D.4), it does not have a strict formula and we believe scoring rules are enough.

---

### Official Review · Reviewer_uFwV · 2023-08-05

**Soundness:** 3

**Excitement:**

3: Ambivalent: It has merits (e.g., it reports state-of-the-art results, the idea is nice), but there are key weaknesses (e.g., it describes incremental work), and it can significantly benefit from another round of revision. However, I won't object to accepting it if my co-reviewers champion it.

**Paper Topic And Main Contributions:**

This paper primarily focuses on evaluating conversational recommender systems (CRSs) using large language models (LLMs).

Its main contributions involve the exploration of ChatGPT's performance in conversational recommendation tasks and the proposal of a method that employs large language model to construct a user simulator for evaluating CRSs performance through simulated dialogues.

However, the proposed method iEvaLM seems less practical in the research. ChatGPT's performance is inconsistent[1], raising concerns about ensuring consistent evaluations from the LLM-based user simulator.

[1] Lingjiao Chen, Matei Zaharia, James Zou, How Is ChatGPT’s Behavior Changing over Time?, https://arxiv.org/pdf/2307.09009.pdf

**Questions For The Authors:**

1. How do you ensure the reproducibility of results? Given that ChatGPT is not entirely open-source and undergoes frequent updates, and considering recent findings [1] suggesting a significant performance drop compared to versions from a few months ago, how can you guarantee the reproducibility of your approach that heavily relies on ChatGPT?
2. What does "LLM-based scorer" refers to in lines 371-372? How was it implemented?
3. When testing ChatGPT, did you provide a candidate movie list? If not, how did you ensure that the recommended movies fall within the evaluation scope? For instance, if the target movie is "The Avengers," but ChatGPT recommends "The Avengers 4."

[1] Lingjiao Chen, Matei Zaharia, James Zou, "How Is ChatGPT’s Behavior Changing over Time?" https://arxiv.org/pdf/2307.09009.pdf

**Reasons To Accept:**

1. This paper investigates ChatGPT's capabilities in completing conversational recommender tasks on two datasets.
2. This paper introduces an interactive approach using LLM-based user simulators to evaluate CRSs.
3. This paper is well written and organized.

**Reasons To Reject:**

1. This paper focusing solely on ChatGPT's capacities in conversational recommendation tasks. It would be valuable to explore the performance of other open-source large language models in conversational recommendation settings. Does the number of parameters such as 7B, 13B, or even larger affect the findings? Providing demonstrations or examples to ChatGPT might be beneficial, such as 1-shot, 3-shot scenarios, considering that experimental results are obtained on a 0-shot scenario.
2. The reasons behind ChatGPT's poor performance in conversational recommender tasks lack evidence.

- Line 303. Traditional CRSs are also limited by explicit user preferences. This may be why ChatGPT does not do well. But the analysis in lines 313-320 does not seem to substantiate this claim convincingly.
- Line 321. I don't think this is the reason why ChatGPT doesn't do well with conversational recommendation tasks. Traditional datasets are manually annotated with fixed settings and rules. So without giving ChatGPT examples, it is likely that it will not be able to follow the patterns of the conversations in the dataset to make recommendations.

3. The paper lacks theoretical support for the reliability of the evaluation method. In dialogue evaluation-related research, correlation coefficients are often computed to measure the effectiveness of an evaluation method. I don't think the experiments carried out in the paper show that iEvaLM is objective and fair. Moreover, the reproducibility of iEvaLM's evaluation results also needs to be ensured as an evaluation method.

**Reproducibility:**

4: Could mostly reproduce the results, but there may be some variation because of sample variance or minor variations in their interpretation of the protocol or method.

**Reviewer Confidence:**

4: Quite sure. I tried to check the important points carefully. It's unlikely, though conceivable, that I missed something that should affect my ratings.

---

> ### Author Rebuttal · Authors · 2023-08-29
>
> We would like to express our gratitude for reviewing our paper and providing valuable feedback. We have carefully considered your comments and have addressed each of your concerns below.
>
> **1. How do you ensure the reproducibility of results? Given that ChatGPT is not entirely open-source and undergoes frequent updates, and considering recent findings [1] suggesting a significant performance drop compared to versions from a few months ago, how can you guarantee the reproducibility of your approach that heavily relies on ChatGPT?**
>
> We appreciate your concern regarding the reproducibility of our results related to ChatGPT. We have tried our best to ensure the reproducibility of the experimental results.
>
> - As stated in Section 2.2, lines 219-224, we have followed existing work [1, 2] to utilize the publicly available model gpt-3.5-turbo-0301 provided in the OpenAI API instead of the web version, which does not receive updates.
>
> - Additionally, we have set the temperature to 0 for gpt-3.5-turbo-0301, as suggested by OpenAI, to ensure the most deterministic generation.
>
> Moreover, we have provided all of our code and data via the anonymous link in the paper. Reviewers P19e and ANzj also gave high scores for the reproducibility of our work.
>
> [1] Md Tahmid Rahman Laskar, M Saiful Bari, Mizanur Rahman, Md Amran Hossen Bhuiyan, Shafiq Joty, Jimmy Huang. A Systematic Study and Comprehensive Evaluation of ChatGPT on Benchmark Datasets. ACL 2023.
>
> [2] Weiwei Sun, Lingyong Yan, Xinyu Ma, Pengjie Ren, Dawei Yin, Zhaochun Ren. Is ChatGPT Good at Search? Investigating Large Language Models as Re-Ranking Agent. arxiv 2023.
>
> **2. What does "LLM-based scorer" refers to in lines 371-372? How was it implemented?**
>
> We have introduced the LLM-based scorer in Section 4.4, lines 430-443. The LLM-based scorer is proposed to reduce the need for human scoring of the subjective metric persuasiveness. It was implemented using the text-davinci-003 model provided in the OpenAI API. For a detailed description of the definition and implementation of the LLM-based scorer, please refer to Section 4.4, lines 430-443.
>
> **3. When testing ChatGPT, did you provide a candidate movie list? If not, how did you ensure that the recommended movies fall within the evaluation scope? For instance, if the target movie is "The Avengers," but ChatGPT recommends "The Avengers 4."**
>
> Thanks for the comment! We do not provide a candidate movie list when testing ChatGPT since it has been shown that ChatGPT can exhibit unexpected behaviors like repetition and missing when required to rank a list [1].
>
> Considering this, in Section 3.1 ("Integrating Recommendation Models"), we follow existing work [2] to utilize external recommendation models to constrain the output space. We concatenate the conversation history and responses generated by ChatGPT as inputs for these recommendation models. We consider two models: MESE [3] and text-embedding-ada-002.
>
> - MESE is a classification-based CRS model trained with the training set of the dataset for evaluation, so its predictions will definitely fall within the evaluation scope.
>
> - Text-embedding-ada-002 is a model from the OpenAI API that can be used to translate item descriptions into embeddings. Then, the recommendation can be made through calculating the similarity between dialogue context and candidate items. We constrain the candidate items to those from the dataset for evaluation; thus, the matched items definitely fall within the evaluation scope.
>
> Therefore, the recommended items will definitely fall within the evaluation scope, and the approach we take can avoid the pitfalls of providing a candidate movie list.
>
> [1] Weiwei Sun, Lingyong Yan, Xinyu Ma, Pengjie Ren, Dawei Yin, Zhaochun Ren. Is ChatGPT Good at Search? Investigating Large Language Models as Re-Ranking Agent. arxiv 2023.
>
> [2] Junling Liu, Chao Liu, Peilin Zhou, Renjie Lv, Kang Zhou, Yan Zhang. Is ChatGPT a Good Recommender? A Preliminary Study. arxiv 2023.
>
> [3] Bowen Yang, Cong Han, Yu Li, Lei Zuo, Zhou Yu. Improving Conversational Recommendation Systems' Quality with Context-Aware Item Meta Information. NAACL 2022.
>
> **4. This paper focusing solely on ChatGPT's capacities in conversational recommendation tasks. It would be valuable to explore the performance of other open-source large language models in conversational recommendation settings. Does the number of parameters such as 7B, 13B affect the findings? Providing demonstrations or examples to ChatGPT might be beneficial, such as 1-shot, 3-shot scenarios, considering that experimental results are obtained on a 0-shot scenario.**
>
> Thanks for the comment! Since this was an initial investigation into using LLMs for CRS, we focused specifically on ChatGPT, which is believed to be the most powerful LLM that is publicly available when submitting the paper (GPT-4 requires an application).
>
> We appreciate your suggestion to explore other LLMs with different scales and the few-shot setting for ChatGPT. However, you may notice that our paper has very tight page space, with extensive analysis and experiments. Since the results of other LLMs and the few-shot setting (as shown below) do not affect the rethinking of the traditional evaluation approach, we have to keep the most important model, ChatGPT, as shown in Table 2.
>
> Here, as requested by the reviewer, we report the evaluation results of Vicuna 7B and 13B, Baize 7B and 13B (we did not have enough resources to test larger models like 33B), and the 1-shot and 3-shot scenario for ChatGPT.
>
> - ReDial
>   - Open-source LLM Vicuna 7B and 13B, Baize 7B and 13B
>   \\begin{array} {|l|c|c|}
>   \\hline
>   \\textrm{Models} & \\textrm{Recall@1} & \\textrm{Recall@10} \\\\
>   \\hline
>   \\textrm{ChatGPT} & 0.034 & 0.172 \\\\
>   \\hline
>   \\textrm{+ MESE} & 0.040 & 0.198 \\\\
>   \\hline
>   \\textrm{+ text-embedding-ada-002} & 0.058 & 0.179 \\\\
>   \\hline
>   \\textrm{Vicuna-7B} & 0.025 & 0.138 \\\\
>   \\hline
>   \\textrm{+ MESE} & 0.039 & 0.193 \\\\
>   \\hline
>   \\textrm{+ text-embedding-ada-002} & 0.042 & 0.131 \\\\
>   \\hline
>   \\textrm{Vicuna-13B} & 0.032 & 0.161 \\\\
>   \\hline
>   \\textrm{+ MESE} & 0.037 & 0.189 \\\\
>   \\hline
>   \\textrm{+ text-embedding-ada-002} & 0.046 & 0.127 \\\\
>   \\hline
>   \\textrm{Baize-7B} & 0.034 & 0.158 \\\\
>   \\hline
>   \\textrm{+ MESE} & 0.028 & 0.144 \\\\
>   \\hline
>   \\textrm{+ text-embedding-ada-002} & 0.045 & 0.134 \\\\
>   \\hline
>   \\textrm{Baize-13B} & 0.030 & 0.167 \\\\
>   \\hline
>   \\textrm{+ MESE} & 0.027 & 0.140 \\\\
>   \\hline
>   \\textrm{+ text-embedding-ada-002} & 0.046 & 0.131 \\\\
>   \\hline
>   \\end{array}
>   - 1-shot and 3-shot for ChatGPT
>   \\begin{array} {|l|c|c|}
>   \\hline
>   \\textrm{Models} & \\textrm{Recall@1} & \\textrm{Recall@10} \\\\
>   \\hline
>   \\textrm{(0-shot) ChatGPT} & 0.034 & 0.172 \\\\
>   \\hline
>   \\textrm{(0-shot) ChatGPT + MESE} & 0.040 & 0.198 \\\\
>   \\hline
>   \\textrm{(0-shot) ChatGPT + text-embedding-ada-002} & 0.058 & 0.179 \\\\
>   \\hline
>   \\textrm{(1-shot) ChatGPT} & 0.031 & 0.148 \\\\
>   \\hline
>   \\textrm{(1-shot) ChatGPT + MESE} & 0.038 & 0.190 \\\\
>   \\hline
>   \\textrm{(1-shot) ChatGPT + text-embedding-ada-002} & 0.058 & 0.165 \\\\
>   \\hline
>   \\textrm{(3-shot) ChatGPT} & 0.037 & 0.176 \\\\
>   \\hline
>   \\textrm{(3-shot) ChatGPT + MESE} & 0.038 & 0.195 \\\\
>   \\hline
>   \\textrm{(3-shot) ChatGPT + text-embedding-ada-002} & 0.061 & 0.174 \\\\
>   \\hline
>   \\end{array}
>
> - OpenDialKG
>   - Open-source LLM Vicuna 7B and 13B, Baize 7B and 13B
>   \\begin{array} {|l|c|c|}
>   \\hline
>   \\textrm{Models} & \\textrm{Recall@1} & \\textrm{Recall@10} \\\\
>   \\hline
>   \\textrm{ChatGPT} & 0.105 & 0.264 \\\\
>   \\hline
>   \\textrm{+ MESE} & 0.258 & 0.531 \\\\
>   \\hline
>   \\textrm{+ text-embedding-ada-002} & 0.370 & 0.633 \\\\
>   \\hline
>   \\textrm{Vicuna-7B} & 0.237 & 0.437 \\\\
>   \\hline
>   \\textrm{+ MESE} & 0.245 & 0.523 \\\\
>   \\hline
>   \\textrm{+ text-embedding-ada-002} & 0.332 & 0.579 \\\\
>   \\hline
>   \\textrm{Vicuna-13B} & 0.221 & 0.438 \\\\
>   \\hline
>   \\textrm{+ MESE} & 0.246 & 0.527 \\\\
>   \\hline
>   \\textrm{+ text-embedding-ada-002} & 0.346 & 0.588 \\\\
>   \\hline
>   \\textrm{Baize-7B} & 0.184 & 0.397 \\\\
>   \\hline
>   \\textrm{+ MESE} & 0.229 & 0.530 \\\\
>   \\hline
>   \\textrm{+ text-embedding-ada-002} & 0.316 & 0.572 \\\\
>   \\hline
>   \\textrm{Baize-13B} & 0.235 & 0.440 \\\\
>   \\hline
>   \\textrm{+ MESE} & 0.240 & 0.508 \\\\
>   \\hline
>   \\textrm{+ text-embedding-ada-002} & 0.311 & 0.545 \\\\
>   \\hline
>   \\end{array}
>   - 1-shot and 3-shot for ChatGPT
>   \\begin{array} {|l|c|c|}
>   \\hline
>   \\textrm{Models} & \\textrm{Recall@1} & \\textrm{Recall@10} \\\\
>   \\hline
>   \\textrm{(0-shot) ChatGPT} & 0.105 & 0.264 \\\\
>   \\hline
>   \\textrm{(0-shot) ChatGPT + MESE} & 0.258 & 0.531 \\\\
>   \\hline
>   \\textrm{(0-shot) ChatGPT + text-embedding-ada-002} & 0.370 & 0.633 \\\\
>   \\hline
>   \\textrm{(1-shot) ChatGPT} & 0.077 & 0.211 \\\\
>   \\hline
>   \\textrm{(1-shot) ChatGPT + MESE} & 0.212 & 0.487 \\\\
>   \\hline
>   \\textrm{(1-shot) ChatGPT + text-embedding-ada-002} & 0.374 & 0.648 \\\\
>   \\hline
>   \\textrm{(3-shot) ChatGPT} & 0.100 & 0.227 \\\\
>   \\hline
>   \\textrm{(3-shot) ChatGPT + MESE} & 0.255 & 0.523 \\\\
>   \\hline
>   \\textrm{(3-shot) ChatGPT + text-embedding-ada-002} & 0.374 & 0.651 \\\\
>   \\hline
>   \\end{array}
>
> As you can see, the performance of Vicuna, Baize, and few-shot ChatGPT also falls behind the top-performing method. Therefore, these results do not affect the rethinking of the traditional evaluation approach.
>
> However, we believe it would be valuable to explore other LLMs and settings. So in the revised version, we will move some analysis to the appendix and expand Table 2 to include these additional results.
>
> **5. The reasons behind ChatGPT's poor performance in conversational recommender tasks lack evidence.**
> - **Line 303. Traditional CRSs are also limited by explicit user preferences. This may be why ChatGPT does not do well. But the analysis in lines 313-320 does not seem to substantiate this claim convincingly.**
>
> Thanks for the insightful comment! You correctly mentioned that traditional CRSs are also limited by explicit user preferences. However, this issue is actually more serious for ChatGPT.
>
> As shown in existing work [1, 2], benchmark datasets like ReDial have a long-tail distribution over items (Figure 2(a) in [1]), and CRSs suffer from popularity bias [2]. Since traditional CRSs are obtained from supervised training, these issues lead to unreasonably high scores for examples whose labels are items with high frequency (Figure 3(a) in [1]), even if these examples lack explicit user preference.
>
> In contrast, ChatGPT is not fine-tuned in our evaluation. Therefore, for examples without explicit user preference, ChatGPT cannot utilize the item distributions of the datasets like traditional CRSs to generate recommendations.
>
> Consequently, lack of explicit user preference may not be very serious for traditional CRSs but is crucial for the failure of ChatGPT.
>
> [1] Lingzhi Wang, Huang Hu, Lei Sha, Can Xu, Kam-Fai Wong, Daxin Jiang. RecInDial: A Unified Framework for Conversational Recommendation with Pretrained Language Models. AACL 2022.
>
> [2] Allen Lin, Jianling Wang, Ziwei Zhu, James Caverlee. Quantifying and Mitigating Popularity Bias in Conversational Recommender Systems. CIKM 2022.
>
> - **Line 321. I don't think this is the reason why ChatGPT doesn't do well with conversational recommendation tasks. Traditional datasets are manually annotated with fixed settings and rules. So without giving ChatGPT examples, it is likely that it will not be able to follow the patterns of the conversations in the dataset to make recommendations.**
>
> Thanks for the comment! We agree with you that traditional datasets are annotated with fixed settings and rules. Actually, our approach is seamlessly integrated with these datasets, which keeps the original labels and takes them as the preference of simulated users. However, we would like to clarify that the key problem is not following the patterns but the limited information about the user's preference.
>
> As shown in the example below, even if ChatGPT follows the patterns to generate a response with the intention of recommendation, it only summarizes the user's preference in the dialogue history. However, since there are so many items that fit the requirement ("musical movie"), it is still challenging to provide accurate recommendations.
>
> \\begin{array} {|l|l|l|}
> \\hline
> & \\textrm{Seeker} & \\textrm{I like The Cabin in the Woods. do you have it or something similar}
> \\newline
> \\textrm{Dialogue history} & \\textrm{Recommender} & \\textrm{That's a great movie. Do you like Kristen Connolly? She starred in Cabin in the Woods.}
> \\newline
> & \\textrm{Seeker} & \\textrm{actually I like something like High School Musical 2.} \\\\
> \\hline
> \\textrm{Response by ChatGPT} & \\textrm{Recommender} & \\textrm{If you enjoyed High School Musical 2, you might like other musical movies.} \\\\
> \\hline
> \\hline
> \\end{array}
>
> Furthermore, as the reviewer suggested in Q4, we tested the performance in the 1-shot and 3-shot scenarios using responses with the intention of recommendation as demonstrations. You can refer to Q4 for the performance table. It can observed that the performance of ChatGPT in the few-shot setting is comparable with the zero-shot setting and still falls behind the top-performing methods, which further reinforces the importance of the reason "Lack of Proactive Clarification" instead of following the patterns.
>
> **6. The paper lacks theoretical support for the reliability of the evaluation method. In dialogue evaluation-related research, correlation coefficients are often computed to measure the effectiveness of an evaluation method. I don't think the experiments carried out in the paper show that iEvaLM is objective and fair. Moreover, the reproducibility of iEvaLM's evaluation results also needs to be ensured as an evaluation method.**
>
> Thanks for the comment! We acknowledge the importance of correlation coefficients. However, commonly used correlation coefficients in dialogue evaluation-related research, such as Pearson, Spearman, and Kendall's tau, cannot accurately reflect the correlation in our scenario.
>
> As stated in Section 5.2.2, we compare our LLM-based scorer with human annotators to evaluate its reliability for scoring the subjective metric persuasiveness. This metric belongs to ordinal data since its value range is {0, 1, 2} with an inherent order (2 > 1 > 0). However, since Pearson correlation coefficient is used for quantitative data, it cannot be used to effectively measure the correlation. While Spearman and Kendall's tau correlation coefficients are used for ordinal data, they can only measure the monotonic relationship between variables, which does not apply to the comparison between scores of persuasiveness. For example, if the human gives a score of 1 for persuasiveness, giving a score of 0 and 2 are equally bad for the LLM-based scorer, only 2 is good. Our explanation is also supported by the discussion on using correlation measures to evaluate NLG metrics [1].
>
> In fact, we have tried using these metrics but obtained abnormal values. As shown in the example below (taking from our collected comparison data), it is evident that these two variables are correlated (with only 14 differences out of 100 observations). However, the values of these coefficients indicate that they are not correlated at all (very close to 0). This discrepancy suggests that these correlation coefficients are not suitable for our scenario.
>
> ```
> import numpy as np
> from scipy import stats
> x = np.array([2.0, 2.0, 1.0, 2.0, 2.0, 2.0, 2.0, 2.0, 2.0, 2.0, 2.0, 2.0, 2.0, 2.0, 2.0, 2.0, 2.0, 2.0, 2.0, 2.0, 1.0, 2.0, 2.0, 2.0, 2.0, 2.0, 2.0, 2.0, 2.0, 2.0, 2.0, 2.0, 2.0, 2.0, 2.0, 0.0, 2.0, 2.0, 2.0, 2.0, 2.0, 2.0, 2.0, 2.0, 2.0, 2.0, 2.0, 2.0, 2.0, 2.0, 2.0, 2.0, 2.0, 2.0, 2.0, 2.0, 2.0, 2.0, 2.0, 2.0, 2.0, 2.0, 2.0, 2.0, 2.0, 2.0, 2.0, 2.0, 2.0, 2.0, 2.0, 2.0, 2.0, 2.0, 2.0, 2.0, 2.0, 2.0, 2.0, 2.0, 2.0, 2.0, 2.0, 2.0, 2.0, 2.0, 2.0, 2.0, 2.0, 2.0, 2.0, 2.0, 2.0, 2.0, 2.0, 2.0, 2.0, 2.0, 2.0, 2.0])
> print(len(x))  # 100
> y = np.array([2.0, 2.0, 2.0, 2.0, 0.0, 2.0, 2.0, 2.0, 2.0, 1.0, 2.0, 2.0, 0.0, 2.0, 2.0, 2.0, 1.0, 2.0, 2.0, 2.0, 2.0, 2.0, 1.0, 2.0, 0.0, 2.0, 2.0, 2.0, 2.0, 2.0, 2.0, 0.0, 2.0, 2.0, 2.0, 2.0, 2.0, 2.0, 1.0, 2.0, 2.0, 2.0, 2.0, 2.0, 2.0, 2.0, 2.0, 2.0, 2.0, 2.0, 2.0, 2.0, 2.0, 2.0, 2.0, 2.0, 2.0, 0.0, 2.0, 2.0, 2.0, 2.0, 1.0, 2.0, 2.0, 2.0, 2.0, 2.0, 2.0, 2.0, 2.0, 2.0, 0.0, 2.0, 2.0, 2.0, 2.0, 2.0, 2.0, 2.0, 2.0, 2.0, 2.0, 2.0, 2.0, 2.0, 2.0, 2.0, 2.0, 2.0, 2.0, 2.0, 2.0, 2.0, 2.0, 2.0, 2.0, 2.0, 2.0, 2.0])
> print(len(y))  # 100
> print((x != y).sum())  # 14 differences
> print(stats.pearsonr(x, y).statistic)  # -0.05506796823644914
> print(stats.spearmanr(x, y).statistic)  # -0.06171640287008385
> print(stats.kendalltau(x, y).statistic)  # -0.06069243286297407
> ```
>
> Considering this, we follow existing work on user simulation in CRS [2] to compare the score distributions between human and LLM-based scorers, as shown in Table 7. The similarity between these two distributions demonstrates the reliability of our LLM-based scorer for the Persuasiveness metric. Therefore, our performance measurement, which consists of Recall and Persuasiveness, is reliable, given that Recall itself is an objective metric.
>
> Building upon this, we further examine the LLM-based user simulator, which is a replacement for humans to interact with the system. We follow the approach outlined in the existing work [2] to compare the performance obtained using the user simulator and humans. As show in Table 8, the ranking obtained from our user simulator is consistent with that of real users, and their absolute scores are also comparable. These results suggest that our user simulator is also reliable.
>
> After evaluating both components, the user simulator, and the performance measurement, we can conclude that iEvaLM provides an objective and fair evaluation of CRS.
>
> Regarding reproducibility, we provide detailed information in Q1. Please refer to that for the response.
>
> [1] Ananya B. Sai, Akash Kumar Mohankumar, Mitesh M. Khapra. A Survey of Evaluation Metrics Used for NLG Systems. ACM Computing Surveys 2022.
>
> [2] Shuo Zhang, Krisztian Balog. Evaluating Conversational Recommender Systems via User Simulation. KDD 2020.
>
> **7. Reproducibility: 1: Could not reproduce the results here no matter how hard they tried.**
>
> We detail this in Q1. Please refer to that for the response.

---

### Official Review · Reviewer_P19e · 2023-08-07

**Soundness:** 3

**Excitement:**

3: Ambivalent: It has merits (e.g., it reports state-of-the-art results, the idea is nice), but there are key weaknesses (e.g., it describes incremental work), and it can significantly benefit from another round of revision. However, I won't object to accepting it if my co-reviewers champion it.

**Paper Topic And Main Contributions:**

The paper is about evaluating conversational recommender systems (CRSs) in the era of large language models (LLMs) such as ChatGPT. The authors first investigate the performance of ChatGPT on existing CRS datasets and find that it is unsatisfactory due to the limitations of the current evaluation protocol. Then, they propose a new interactive evaluation approach, iEvaLM, which leverages LLM-based user simulators to examine the accuracy and explainability of recommendations. Through experiments, they show that ChatGPT significantly improves its performance under iEvaLM and outperforms existing CRSs in both aspects. They also demonstrate the effectiveness and reliability of their evaluation approach. The paper contributes to the understanding and evaluation of LLMs for conversational recommendation.

**Reasons To Accept:**

1. The paper proposes LLMs as user simulators to evaluate the performance of CRS methods, which is reasonable.
2. Several human evaluations have been done to validate the quality of the LLM-based user simulator and consistency with human preferences.
3. The paper is well-written and easy to understand.

**Reasons To Reject:**

1. Some experimental designs are confusing
    1.  GPT-3.5-turbo is used for CRS while text-davinci-003 is used for user simulator.
2. The reasons in Section 3.3 can not fully explain why ChatGPT fails to outperform other methods.
    1. For the first reason, other baseline methods are also unable to collect enough evidences.
    2. For the second reason, I suppose forcing ChatGPT to make recommendation at that step with prompt design can solve the problem.

--------------
These issues are well addressed in the authors' response.

**Reproducibility:**

5: Could easily reproduce the results.

**Reviewer Confidence:**

2: Willing to defend my evaluation, but it is fairly likely that I missed some details, didn't understand some central points, or can't be sure about the novelty of the work.

---

> ### Author Rebuttal · Authors · 2023-08-29
>
> Thank you for taking the time to review our paper. In response to your comments, we have carefully considered each point and addressed them as follows.
>
> **1. Some experimental designs are confusing: GPT-3.5-turbo is used for CRS while text-davinci-003 is used for user simulator.**
>
> Thanks for the comment! For CRS, we use gpt-3.5-turbo since it has acquired a wealth of world knowledge during pre-training and is also specially optimized for conversation, which is expected to be an excellent LLM-based CRS.
>
> For the user simulator, we use text-davinci-003, as it demonstrates a capable ability to follow instructions during multi-turn dialogue, which is what we need for a user simulator with expected behaviors set in Section 4.3. In contrast, gpt-3.5-turbo is specially trained for dialogue, which tends to generate responses more influenced by the dialogue history rather than adhering to the provided instructions.
>
> As shown in the example below, text-davinci-003 perfectly follows the instruction to refuse the recommended items since they are not the target ones in the instruction, while gpt-3.5-turbo starts to chit-chat with the system without following the instruction.
>
> \\begin{array} {|l|l|}
> \\hline
> \\textrm{\textbf{Instruction}} & \\textrm{You are a seeker chatting with a recommender for recommendation. Your target items: "Click (2006)", "It (2017)". You must follow the instructions below during chat.}
> \\newline
> & \\textrm{If the recommender recommends "Click (2006)", "It (2017)", you should accept.}
> \\newline
> & \\textrm{If the recommender recommends other items, you should refuse them and provide the information about "Click (2006)", "It (2017)". You should never directly tell the target item title.}
> \\newline
> & \\textrm{If the recommender asks for your preference, you should provide the information about "Click (2006)", "It (2017)". You should never directly tell the target item title.} \\\\
> \\hline
> \\textrm{\textbf{Recommender}} & \\textrm{Tell me what would you like to watch?} \\\\
> \\hline
> \\textrm{\textbf{Seeker}} & \\textrm{Easy A (2010)} \\\\
> \\hline
> \\textrm{\textbf{Recommender}} & \\textrm{Did you watch Avengers: Infinity War (2018) ? Or The Black Panthers: Vanguard of the Revolution (2015) ? i really liked it} \\\\
> \\hline
> \\textrm{\textbf{Seeker}} & \\textrm{No , I think I would like to watch Easy A (2010) It's funny} \\\\
> \\hline
> \\textrm{\textbf{Recommender}} & \\textrm{Great! Based on your preference for Easy A (2010), here are some movie recommendations for you:}
> \\newline
> & \\textrm{1. The DUFF (2015)}
> \\newline
> & \\textrm{2. Mean Girls (2004)}
> \\newline
> & \\textrm{3. 10 Things I Hate About You (1999)}
> \\newline
> & \\textrm{4. Clueless (1995)}
> \\newline
> & \\textrm{5. Legally Blonde (2001)}
> \\newline
> & \\textrm{6. She's the Man (2006)}
> \\newline
> & \\textrm{7. The Edge of Seventeen (2016)}
> \\newline
> & \\textrm{8. The Perks of Being a Wallflower (2012)}
> \\newline
> & \\textrm{9. Juno (2007)}
> \\newline
> & \\textrm{10. The Fault in Our Stars (2014)}
> \\newline
> & \\textrm{I hope you find these recommendations helpful. Let me know if you need any more suggestions!} \\\\
> \\hline
> \\textrm{\textbf{Seeker (gpt-3.5-turbo)}} & \\textrm{Thanks for the recommendations! I have already watched some of them, but I will definitely check out the ones I haven't seen yet. By the way, have you watched any good movies lately?} \\\\
> \\hline
> \\textrm{\textbf{Seeker (text-davinci-003)}} & \\textrm{These look great, thank you! I'm actually looking for something a bit different though. I'm looking for a comedy and a horror movie. Do you have any recommendations for those genres?} \\\\
> \\hline
> \\end{array}
>
> Furthermore, text-davinci-003 has been shown to outperform GPT-3.5-turbo on the WebShop task [1]. The WebShop task involves an agent purchasing a product based on user instructions through multi-turn interactions with the web. It evaluates the agent's ability to follow instructions in the context of multi-turn interactions. This ability aligns with our needs for the user simulator, and the superior performance of text-davinci-003 further reinforces our experimental designs.
>
> We apologize for any confusion caused by this experimental design and will provide more thorough explanations in the revised version.
>
> [1] Qiantong Xu, Fenglu Hong, Bo Li, Changran Hu, Zhengyu Chen, Jian Zhang. On the Tool Manipulation Capability of Open-source Large Language Models. arxiv 2023.
>
> **2. The reasons in Section 3.3 can not fully explain why ChatGPT fails to outperform other methods.**
>
> - **For the first reason, other baseline methods are also unable to collect enough evidences.**
>
> Thanks for the insightful comment! Regarding the first reason ("Lack of Explicit User Preference"), you correctly mentioned that baseline methods also face the challenge of collecting enough evidence. We agree with your observation; however, this issue is actually more serious for ChatGPT.
>
> Existing work [1, 2] has shown that benchmark datasets like ReDial have a long-tail distribution over items (Figure 2(a) in [1]), and CRSs suffer from popularity bias [2]. Since baseline methods are trained on these datasets, this issue results in unreasonably high scores for examples whose labels are items with high frequency (Figure 3(a) in [1]), even if these examples lack explicit user preference.
>
> In contrast, ChatGPT is not fine-tuned in our evaluation. Therefore, when it comes to examples without explicit user preference, ChatGPT cannot utilize the item distributions of the datasets like baseline methods. As a result, the "Lack of Explicit User Preference" becomes an important reason for the failure of ChatGPT.
>
> [1] Lingzhi Wang, Huang Hu, Lei Sha, Can Xu, Kam-Fai Wong, Daxin Jiang. RecInDial: A Unified Framework for Conversational Recommendation with Pretrained Language Models. AACL 2022.
>
> [2] Allen Lin, Jianling Wang, Ziwei Zhu, James Caverlee. Quantifying and Mitigating Popularity Bias in Conversational Recommender Systems. CIKM 2022.
>
> - **For the second reason, I suppose forcing ChatGPT to make recommendation at that step with prompt design can solve the problem.**
>
> Thanks for the comment! Regarding the second reason ("Lack of Proactive Clarification"), we have considered the method of forcing ChatGPT to make recommendations in our paper, as suggested by the reviewer.
>
> - For directly making recommendations by ChatGPT itself, we consider this approach in Section 3.1 ("Zero-shot prompting"), where ChatGPT is prompted to generate recommendations. As shown in Table 2 (the first line of ChatGPT), such an approach only achieves average performance and falls far behind the top-performing methods.
>
> - For making recommendations by external recommendation models, we consider this approach in Section 3.1 ("Integrating Recommendation Models"), where the response by ChatGPT is used as the input for recommendation models. Here, we would like to clarify that the key problem is not the intention of the response (recommendation or clarification) but the limited information of the user preference in the dialogue context.
>
>   - As shown in the example below, even if ChatGPT generates a response with the intention of recommendation, it only summarizes the user preference in the dialogue history. However, since so many items fit the requirement ("musical movies"), it is still hard to give accurate recommendations.
>
>     \\begin{array} {|l|l|l|}
>     \\hline
>     & \\textrm{Seeker} & \\textrm{I like The Cabin in the Woods. do you have it or something   similar}
>     \\newline
>     \\textrm{Dialogue history} & \\textrm{Recommender} & \\textrm{That's a great movie. Do you like Kristen Connolly? She starred in Cabin in the Woods.}
>     \\newline
>     & \\textrm{Seeker} & \\textrm{actually I like something like High School Musical 2.} \\\\
>     \\hline
>     \\textrm{Response by ChatGPT} & \\textrm{Recommender} & \\textrm{If you enjoyed High School Musical 2, you might like other musical movies.} \\\\
>     \\hline
>     \\hline
>     \\end{array}
>
>   - Instead, our evaluation approach, iEvaLM, allows further interaction between the user and system, where ChatGPT can gather the necessary information to differentiate among candidates. Table 5 demonstrates the significant improvement of baseline methods and ChatGPT in iEvaLM, further justifying the second reason ("Lack of Proactive Clarification").
>
> We have provided sufficient explanations for the two mentioned reasons for the failure of ChatGPT and promise to add more thorough explanations to fully address your concerns in the revised version.

---

### Official Review · Reviewer_UZ7K · 2023-08-13

**Soundness:** 2

**Excitement:**

3: Ambivalent: It has merits (e.g., it reports state-of-the-art results, the idea is nice), but there are key weaknesses (e.g., it describes incremental work), and it can significantly benefit from another round of revision. However, I won't object to accepting it if my co-reviewers champion it.

**Paper Topic And Main Contributions:**

To address the limitations of existing evaluation protocols, which might overemphasize matching with ground-truth items and overlook the interactive nature of CRSs, this paper introduces iEvaLM, which leverages LLM-based user simulators.

**Reasons To Accept:**

**Strengths:**

1. This paper addresses an important problem in a timely manner. Evaluation is vital in research, especially at a time when LLM is widely used in real-life applications. Traditional evaluation protocols, like matching with ground-truth items, cannot effectively measure the performance of current CRSs.
2. The proposed approach of adopting LLMs as user simulators for evaluating CRSs is intriguing.
3. This paper conducts a systematic analysis of ChatGPT’s capabilities in conversational recommendation and highlights its limitations under traditional evaluation methods.

**Reasons To Reject:**

**Weaknesses：**

1. Since the user simulator is based on text-davince-003, will iEvaLM be biased towards ChatGPT during evaluation?
2. The robustness of the evaluation framework is unclear. Minor changes in prompts could lead to significant differences in results.
3. Even though the code is open-sourced, there are concerns about the transparency of the evaluation framework given that text-davince-003 is a closed API.

**Reproducibility:**

2: Would be hard pressed to reproduce the results. The contribution depends on data that are simply not available outside the author's institution or consortium; not enough details are provided.

**Reviewer Confidence:**

3: Pretty sure, but there's a chance I missed something. Although I have a good feel for this area in general, I did not carefully check the paper's details, e.g., the math, experimental design, or novelty.

---

> ### Author Rebuttal · Authors · 2023-08-29
>
> We would like to express our gratitude for your valuable feedback on our paper. We have carefully considered your comments and concerns, and we provide our response below to address each of them.
>
> **1. Since the user simulator is based on text-davince-003, will iEvaLM be biased towards ChatGPT during evaluation?**
>
> We appreciate your concern regarding bias in the evaluation process. However, we would like to clarify that the user simulator is not designed to score CRSs; its purpose is solely to interact with the system, while performance is assessed using metrics such as Recall based on the interaction. Therefore, selecting text-davinci-003 as the user simulator will not introduce bias towards ChatGPT during the evaluation.
>
> **2. The robustness of the evaluation framework is unclear. Minor changes in prompts could lead to significant differences in results.**
>
> Thanks for the insightful comment! Robustness is a widespread concern for the evaluation of LLMs since prompts for LLMs exist in the space of natural language. We follow common practice (e.g., Open LLM Leaderboard by HuggingFace) to report the evaluation results using a standard prompt for all models. However, in consideration of robustness, we have also tested some prompts with minor changes as the reviewer suggested.
>
> \\begin{array} {|l|l|}
> \\hline
> \\textrm{Prompt No.} & \\textrm{Prompt for the user simulator} \\\\
> \\hline
> 1 (\\textrm{we show it in the paper}) & \\textrm{You are a seeker chatting with a recommender for recommendation. Your target items: \\{\\textit{target items}\\}. You must follow the instructions below during chat.}
> \\newline
> & \\textrm{If the recommender recommends \\{\\textit{target items}\\}, you should accept.}
> \\newline
> & \\textrm{If the recommender recommends other items, you should refuse them and provide the information about \\{\\textit{target items}\\}. You should never directly tell the target item title.}
> \\newline
> & \\textrm{If the recommender asks for your preference, you should provide the information about \\{\\textit{target items}\\}. You should never directly tell the target item title.} \\\\
> \\hline
> 2 (\\textrm{add a number before each rule}) & \\textrm{You are a seeker chatting with a recommender for recommendation. Your target items: \\{\\textit{target items}\\}. You must follow the instructions below during chat.}
> \\newline
> & \\textrm{1. If the recommender recommends \\{\\textit{target items}\\}, you should accept.}
> \\newline
> & \\textrm{2. If the recommender recommends other items, you should refuse them and provide the information about \\{\\textit{target items}\\}. You should never directly tell the target item title.}
> \\newline
> & \\textrm{3. If the recommender asks for your preference, you should provide the information about \\{\\textit{target items}\\}. You should never directly tell the target item title.} \\\\
> \\hline
> 3 (\\textrm{paraphrase the first sentence}) & \\textrm{You are inquiring with a recommender for recommendations. Your target items: \\{\\textit{target items}\\}. You must follow the instructions below during chat.}
> \\newline
> & \\textrm{If the recommender recommends \\{\\textit{target items}\\}, you should accept.}
> \\newline
> & \\textrm{If the recommender recommends other items, you should refuse them and provide the information about \\{\\textit{target items}\\}. You should never directly tell the target item title.}
> \\newline
> & \\textrm{If the recommender asks for your preference, you should provide the information about \\{\\textit{target items}\\}. You should never directly tell the target item title.} \\\\
> \\hline
> 4 (\\textrm{remove ``You must''}) & \\textrm{You are a seeker chatting with a recommender for recommendation. Your target items: \\{\\textit{target items}\\}. Follow the instructions below during chat.}
> \\newline
> & \\textrm{If the recommender recommends \\{\\textit{target items}\\}, you should accept.}
> \\newline
> & \\textrm{If the recommender recommends other items, you should refuse them and provide the information about \\{\\textit{target items}\\}. You should never directly tell the target item title.}
> \\newline
> & \\textrm{If the recommender asks for your preference, you should provide the information about \\{\\textit{target items}\\}. You should never directly tell the target item title.} \\\\
> \\hline
> \\end{array}
>
> As shown below, there are no significant differences in the evaluation results, and the ranking of models remains consistent across different prompts.
>
> - OpenDialKG (Since requiring too many items at once can sometimes be refused by ChatGPT, following our paper, we only assess Recall@1 and Recall@10 for it)
> \\begin{array} {|c|c|c|c|c|c|}
> \\hline
> \\textrm{Metric} & \\textrm{Prompt No.} & \\textrm{KBRD} & \\textrm{BARCOR} & \\textrm{UniCRS} & \\textrm{ChatGPT} \\\\
> \\hline
> \\textrm{Recall@1} & 1 & 0.234 & 0.314 & 0.314 & 0.400
> \\newline
> & 2 & 0.232 & 0.314 & 0.313 & 0.410
> \\newline
> & 3 & 0.232 & 0.313 & 0.321 & 0.414
> \\newline
> & 4 & 0.234 & 0.313 & 0.320 & 0.409 \\\\
> \\hline
> \\textrm{Recall@10} & 1 & 0.431 & 0.458 & 0.538 & 0.715
> \\newline
> & 2 & 0.428 & 0.458 & 0.536 & 0.716
> \\newline
> & 3 & 0.434 & 0.461 & 0.547 & 0.727
> \\newline
> & 4 & 0.433 & 0.458 & 0.545 & 0.726 \\\\
> \\hline
> \\textrm{Recall@50} & 1 & 0.509 & 0.530 & 0.609 & -
> \\newline
> & 2 & 0.508 & 0.528 & 0.611 & -
> \\newline
> & 3 & 0.517 & 0.529 & 0.618 & -
> \\newline
> & 4 & 0.511 & 0.527 & 0.617 & - \\\\
> \\hline
> \\end{array}
>
> - ReDial (Since requiring too many items at once can sometimes be refused by ChatGPT, following our paper, we only assess Recall@1 and Recall@10 for it)
> \\begin{array} {|c|c|c|c|c|c|}
> \\hline
> \\textrm{Metric} & \\textrm{Prompt No.} & \\textrm{KBRD} & \\textrm{BARCOR} & \\textrm{UniCRS} & \\textrm{ChatGPT} \\\\
> \\hline
> \\textrm{Recall@1} & 1 & 0.035 & 0.034 & 0.107 & 0.146
> \\newline
> & 2 & 0.032 & 0.033 & 0.112 & 0.144
> \\newline
> & 3 & 0.037 & 0.034 & 0.121 & 0.150
> \\newline
> & 4 & 0.035 & 0.035 & 0.124 & 0.149 \\\\
> \\hline
> \\textrm{Recall@10} & 1 & 0.198 & 0.190 & 0.317 & 0.440
> \\newline
> & 2 & 0.198 & 0.188 & 0.319 & 0.430
> \\newline
> & 3 & 0.207 & 0.185 & 0.325 & 0.450
> \\newline
> & 4 & 0.206 & 0.188 & 0.322 & 0.445 \\\\
> \\hline
> \\textrm{Recall@50} & 1 & 0.453 & 0.467 & 0.602 & -
> \\newline
> & 2 & 0.455 & 0.465 & 0.600 & -
> \\newline
> & 3 & 0.460 & 0.462 & 0.606 & -
> \\newline
> & 4 & 0.470 & 0.468 & 0.610 & - \\\\
> \\hline
> \\end{array}
>
> Additionally, you may notice that our paper has limited page space, which necessitates focusing on the most important parts. Given that the robustness of evaluation regarding LLMs is still an open problem without standard and reliable evaluation approaches, we do not include this aspect in our submission. We promise to add a discussion about it in the revised version.
>
> **3. Even though the code is open-sourced, there are concerns about the transparency of the evaluation framework given that text-davince-003 is a closed API.**
>
> We appreciate your concern regarding the transparency of the evaluation framework. We chose text-davinci-003 to implement the user simulator in our work due to its powerful capacity for human alignment. As shown in the evaluation of model-based evaluators from AlpacaEval [1], text-davinci-003 performs much better than ChatGPT in agreement with human evaluators, only falling behind Claude and GPT-4. Since we do not have access to Claude and GPT-4 (as they require application at the time of submission), text-davinci-003 is the best model we can choose for the user simulator. Such a powerful capacity for human alignment is important for simulating realistic dialogues to evaluate the performance of CRSs in close-to-real scenarios.
>
> In addition, the user simulator is seamlessly integrated with public CRS datasets. The preference of the user simulator is set to the ground-truth items from datasets, which are sourced from real human annotators and publicly available.
>
> While it is true that text-davinci-003 is a closed API, we would like to clarify that the evaluation framework itself is independent of any specific API and can be adapted to any model with capable conversational abilities. Here, we report the evaluation results obtained by using the open-sourced model Vicuna-7B to implement the user simulator. It can be observed that the ranking of models remains consistent between text-davinci-003 and Vicuna-7B.
>
> - ReDial (Since requiring too many items at once can sometimes be refused by ChatGPT, following our paper, we only assess Recall@1 and Recall@10 for it)
> \\begin{array} {|c|l|c|c|c|c|}
> \\hline
> \\textrm{User simulator} & \\textrm{Metric} & \\textrm{KBRD} & \\textrm{BARCOR} & \\textrm{UniCRS} & \\textrm{ChatGPT} \\\\
> \\hline
> & \\textrm{Recall@1} & 0.035 & 0.034 & 0.107 & 0.146
> \\newline
> \\textrm{text-davinci-003} & \\textrm{Recall@10} & 0.198 & 0.190 & 0.317 & 0.440
> \\newline
> & \\textrm{Recall@50} & 0.453 & 0.467 & 0.602 & - \\\\
> \\hline
> & \\textrm{Recall@1} & 0.023 & 0.024 & 0.097 & 0.117
> \\newline
> \\textrm{Vicuna} & \\textrm{Recall@10} & 0.167 & 0.166 & 0.298 & 0.419
> \\newline
> & \\textrm{Recall@50} & 0.401 & 0.425 & 0.578 & - \\\\
> \\hline
> \\end{array}
>
> - OpenDialKG (Since requiring too many items at once can sometimes be refused by ChatGPT, following our paper, we only assess Recall@1 and Recall@10 for it)
> \\begin{array} {|c|l|c|c|c|c|}
> \\hline
> \\textrm{User simulator} & \\textrm{Metric} & \\textrm{KBRD} & \\textrm{BARCOR} & \\textrm{UniCRS} & \\textrm{ChatGPT} \\\\
> \\hline
> & \\textrm{Recall@1} & 0.234 & 0.314 & 0.314 & 0.400
> \\newline
> \\textrm{text-davinci-003} & \\textrm{Recall@10} & 0.431 & 0.458 & 0.538 & 0.715
> \\newline
> & \\textrm{Recall@50} & 0.509 & 0.530 & 0.609 & - \\\\
> \\hline
> & \\textrm{Recall@1} & 0.228 & 0.302 & 0.309 & 0.383
> \\newline
> \\textrm{Vicuna} & \\textrm{Recall@10} & 0.425 & 0.441 & 0.530 & 0.689
> \\newline
> & \\textrm{Recall@50} & 0.495 & 0.527 & 0.592 & - \\\\
> \\hline
> \\end{array}
>
> Furthermore, we have made our best efforts to ensure the transparency and reproducibility of our work.
>
> - We set the temperature to 0 for text-davinci-003, as suggested by OpenAI, to make the generation mostly deterministic.
> - We have open-sourced the code, which includes the implementation of the evaluation framework. Researchers can utilize our code, replace text-davinci-003 with open-sourced conversation models, and replicate the evaluation process, thereby promoting transparency and enabling further research in this domain.
>
> [1] Xuechen Li, Tianyi Zhang, Yann Dubois, Rohan Taori, Ishaan Gulrajani, Carlos Guestrin, Percy Liang, Tatsunori B. Hashimoto. AlpacaEval: An Automatic Evaluator of Instruction-following Models. https://github.com/tatsu-lab/alpaca_eval#evaluators
>
> **4. Reproducibility: 2: would be hard pressed to reproduce the results.**
>
> We would like to clarify that, as stated in the response to Q3, we have open-sourced all of our code and data. The evaluation framework itself is independent of any specific API or model. Furthermore, we have followed OpenAI's suggestion to set the temperature to 0 for all the provided APIs, which makes the generation mostly deterministic. Therefore, you can easily reproduce the results following the instructions for running the code. Reviewers P19e and ANzj have also given high scores for the reproducibility of our work.

---

### Meta-Review · Area_Chair_JcZ9 · 2023-09-21

**Recommendation:** 4

**Metareview:**

To address the limitations of existing evaluation protocols, which might overemphasize matching with ground-truth items and overlook the interactive nature of CRSs, this paper introduces iEvaLM, which leverages LLM-based user simulators.

This paper has gone through a thorough discussion among reviewers and many concerns about it have been well addressed. Nonetheless, the paper can still be improved by addressing the following issues:
1. The proposed iEvaLM might be biased towards the LLM used during evaluation.
2. Add a discussion of the potential risks of using iEvaLM, as reviewers have concerns about the objectivity and fairness of the proposed method. It would be nice if the authors could add some use hints or guides for readers or practitioners.

The authors are also encouraged to include all results listed in the rebuttal in their paper if accepted.

---

### Decision · Program_Chairs · 2023-10-07

**Decision:**

Accept-Main

**Comment:**

To address the limitations of existing evaluation protocols, which might overemphasize matching with ground-truth items and overlook the interactive nature of CRSs, this paper introduces iEvaLM, which leverages LLM-based user simulators.

This paper has gone through a thorough discussion among reviewers and many concerns about it have been well addressed. Nonetheless, the paper can still be improved by addressing the following issues:
1. The proposed iEvaLM might be biased towards the LLM used during evaluation.
2. Add a discussion of the potential risks of using iEvaLM, as reviewers have concerns about the objectivity and fairness of the proposed method. It would be nice if the authors could add some use hints or guides for readers or practitioners.

The authors are also encouraged to include all results listed in the rebuttal in their paper if accepted.